# A Systematic Review of Genetic Variants in Glutathione S-Transferase Genes and Their Dual Role in SARS-CoV-2 Pathogenesis: From Acute Respiratory Complications to Long COVID

**DOI:** 10.3390/antiox14080912

**Published:** 2025-07-25

**Authors:** Valeria Villegas Sánchez, Juan Luis Chávez Pacheco, Margarita Isabel Palacios Arreola, Martha Patricia Sierra-Vargas, Luz Adriana Colín Godinez, Víctor Hugo Ahumada Topete, Rosario Fernández Plata, Anjarath Higuera-Iglesias, Roberto Lara-Lemus, Arnoldo Aquino-Gálvez, Luz María Torres-Espíndola, Manuel Castillejos-López

**Affiliations:** 1Escuela Superior de Medicina Programa de Maestría en Ciencias de la Salud, Instituto Politécnico Nacional, Mexico City 11340, Mexico; 32401919513@alumno.enp.unam.mx (V.V.S.); mcoling2300@alumno.ipn.unam.mx (L.A.C.G.); 2Laboratorio de Investigación en Epidemiología e Infectología, Instituto Nacional de Enfermedades Respiratorias “Ismael Cosío Villegas”, Calzada de Tlalpan 4502, Mexico City 14080, Mexico; rosferpla@gmail.com (R.F.P.); higuera.iglesias.anjarath@gmail.com (A.H.-I.); 3Laboratorio de Farmacología, Instituto Nacional de Pediatría, Av. Insurgentes Sur 3700-Letra C, Coyoacán, Mexico City 04530, Mexico; jchavezp@pediatria.gob.mx; 4Departamento de Investigación en Toxicología y Medicina Ambiental, Instituto Nacional de Enfermedades Respiratorias Ismael Cosío Villegas, Secretaría de Salud, Mexico City 14080, Mexico; margarita.palacios@iner.gob.mx (M.I.P.A.); mpsierra@iner.gob.mx (M.P.S.-V.); 5Departamento de Epidemiología Hospitalaria e Infectología, Instituto Nacional de Enfermedades Respiratorias “Ismael Cosío Villegas”, Calzada de Tlalpan 4502, Mexico City 14080, Mexico; victor.ahumada@uehi.mx; 6Departamento de Biomedicina Molecular e Investigación Traslacional, Instituto Nacional de Enfermedades Respiratorias Ismael Cosío Villegas (INER), Mexico City 14080, Mexico; antonio.lara@iner.gob.mx; 7Departamento de Fibrosis Pulmonar, Laboratorio de Biología Molecular, Instituto Nacional de Enfermedades Respiratorias “Ismael Cosío Villegas”, Calzada de Tlalpan 4502, Mexico City 14080, Mexico; arnoldo.aquino@iner.gob.mx; 8Departamento de Bioquímica, Facultad de Medicina, Universidad Nacional Autónoma de México (UNAM), Mexico City 04510, Mexico

**Keywords:** SARS-CoV2, *GST*, long COVID

## Abstract

Oxidative stress (OS) occurs when there is an imbalance between oxidants and antioxidants, leading to disruptions in cellular signaling and causing damage to molecules. Glutathione S-transferase (GST) enzymes are crucial for maintaining redox balance by facilitating glutathione conjugation. Increased oxidative damage has been noted during the COVID-19 pandemic, and its persistence may be linked to the onset of long COVID. This systematic review aimed to assess the relationship between *GST* gene polymorphisms and the prognosis of COVID-19, including long COVID. Adhering to the PRISMA guidelines, a thorough search was carried out in MEDLINE, CENTRAL, PubMed, and EMBASE for studies published from September 2020 to February 2025. Out of an initial selection of 462 articles, ten studies (four concerning COVID-19 severity and six related to long COVID) satisfied the inclusion criteria. The findings regarding *GST* polymorphisms were not consistent, especially concerning the *GSTM1* and *GSTT1* isoforms. Nevertheless, evidence indicates a sustained state of oxidative stress in patients with long COVID. The majority of research has focused on cytosolic GSTs, while the functions of microsomal and mitochondrial GST families remain largely unexplored. These findings suggest that further research into the various *GST* subfamilies and their genetic variants is necessary to enhance our understanding of their impact on COVID-19 severity and the pathophysiology of long COVID.

## 1. Introduction

Oxidative stress (OS) refers to the disproportion between oxidants and antioxidants, tilted in favor of oxidants, which results in disrupted signaling, impaired redox control, and molecular damage [1,2,3]. This condition occurs due to either an increase in the production of reactive oxygen and nitrogen species (RONS) or to a lack of adequate antioxidant defenses [4]. Certain viruses can trigger the production of RONS, resulting in elevated concentrations within the cell (examples include influenza, respiratory syncytial virus, and metapneumovirus) [5]. Low levels of oxidative stress (eustress) are essential for maintaining cellular redox homeostasis [1]. To uphold this equilibrium, both enzymatic and non-enzymatic regulatory mechanisms are present to help neutralize RONS or harmful electrophiles [4,5]. Glutathione (GSH), a non-enzymatic antioxidant tripeptide, primarily plays a critical role in detoxifying xenobiotics [6]. This detoxification process involves the glutathione S-transferases (GST) family of enzymes, which are crucial in phase II reactions [5,6], and also safeguard cells by catalyzing the conjugation of GSH with both endogenous and exogenous electrophilic substances (such as oxidative stress products) [7,8,9,10,11], thereby increasing their water solubility and facilitating renal elimination [12,13,14]. GSTs also participate in cellular signaling and post-translational modifications and contribute to resistance against chemotherapy drugs [14]. They are classified into three families based on their amino acid sequences and substrate specificity: cytosolic, mitochondrial, and microsomal; the cytosolic family is among the most researched, with numerous reports on alterations associated with it [8,9] (Figure 1).

Various polymorphisms of these enzymes have been investigated, including the null genotypes of *GSTM1* and *GSTT1*, which are tied to enzyme inactivation and heightened risk of cellular damage [15].

The detoxification reaction involving GSH is primarily catalyzed by glutathione peroxidase (GPx). The superoxide anion (O_2_^‒^) undergoes dismutation by superoxide dismutase (SOD) to produce hydrogen peroxide (H_2_O_2_), which has two possible fates: one involves iron (Fe^2+^), which through the Fenton reaction generates the hydroxyl radical (HO–), while the other involves interaction with neutrophil myeloperoxidase present in these patients [16], producing hypochlorous acid (HClO–), both potent oxidants necessary for microbial destruction. Compared with O_2_^−^ and HO–, H_2_O_2_ is the least reactive species; it participates in hypoxia signal transduction, cellular differentiation, and proliferation and plays an essential role in the immune response [17]. The generated H_2_O_2_ is metabolized by several antioxidant enzymes, including catalase (CAT), glutathione peroxidase, and thioredoxin peroxidase (peroxiredoxins) [4,8,10,17] (Figure 2).

On 11 March 2020, the World Health Organization (WHO) announced that the coronavirus disease (COVID-19) was a pandemic, with SARS-CoV-2 being the causative agent (as per its English abbreviation) [18]. This illness is associated with a significant rate of morbidity and mortality, primarily because of the occurrence of acute respiratory distress syndrome (ARDS) [19,20], which induces metabolic changes that lead to imbalances, resulting in an increase in OS [21,22,23]. In the acute phase of the disease, there is a rise in reactive oxygen species (ROS) production and oxidative damage prompted by the cytokine storm [22,24], leading to lung injury, respiratory failure, endothelial dysfunction, thrombosis, cardiovascular damage, and multiple organ failure [25,26,27]. It has been documented that a surge in ROS coincides with a depletion of the antioxidant system, showcased by diminished GSH levels in these patients [8,28]. A complication of this disease is long COVID or post-COVID-19 syndrome, which entails the persistence or emergence of new symptoms three months post-infection, lasting for at least two months without an alternative explanation [29,30]. This indicates that OS may be sustained even after the infection’s acute phase.

The severity of COVID-19 is marked by the onset of ARDS, predominantly affecting male patients [14], older individuals [20], and those with chronic degenerative conditions [19,20,31]. An increase in pre-existing oxidative stress linked to these patients’ risk factors, which intensifies the disease’s severity, has been recognized [32]. The aggravation of this redox imbalance may correlate with increased severity and mortality associated with COVID-19 [6]. During infection, the inflammatory response is initiated when the virus attaches to the host cell, with the S protein binding to the angiotensin-converting enzyme 2 (ACE2), inducing a conformational alteration in S1 that reveals cleavage sites for proteolytic enzymes [33,34]. The type 2 transmembrane serine protease (TMPRSS2) present in the host cell facilitates viral entry by cleaving ACE2 and activating the S protein, leading to the fusion of membranes [34]. The virus enters through endocytosis, where it releases its RNA, which associates with the DNA and triggers the replicative cycle [35]. The viral proteins and RNA synthesized assemble into new viral particles, which are expelled from the cell via exocytosis [35,36]. In the context of SARS-CoV-2 infection, angiotensin-converting enzyme (ACE) and ACE2 are key players influencing the development of OS [22]. Both ACE and ACE2 are membrane-anchored enzymes; ACE promotes the conversion of angiotensin I (Ang I) to angiotensin II (Ang II), causing vasoconstriction, inflammation, and oxidative stress through the activation of NADPH oxidase and the production of peroxynitrite anions [6]. Concurrently, ACE2 deactivates Ang I by converting it to angiotensin 1-7 (Ang 1-7), which yields effects contrary to those produced by Ang II [37]. Since SARS-CoV-2 negatively affects the abundance of ACE2 on cell surfaces, the result is OS resulting from the toxic overaccumulation of angiotensin II produced by ACE, leading to an intensified inflammatory response and acute respiratory distress syndrome [6,24]. (Figure 3). The purpose of this article is to conduct a systematic review of the existing literature regarding genetic variants in the *glutathione S-transferase* family of genes. Additionally, it aims to evaluate how these variants influence the severity of COVID-19 and contribute to the development of post-COVID-19 syndrome, commonly referred to as long COVID.

## 2. Materials and Methods

This review was carried out following the guidelines set forth by the Preferred Reporting Items for Systematic Reviews and Meta-Analyses (PRISMA) 2020 (Appendix A). The review started on 1 February 2025 and concluded on 1 July 2025. Articles published between September 2020 and February 2025 that examine *GST* polymorphisms in individuals with COVID-19 or long COVID were included through a computerized literature search conducted using the MEDLINE database (National Library of Medicine, Bethesda, MD), the Cochrane Central Register of Controlled Trials (CENTRAL), PubMed, and EMBASE. The following combinations of search terms were employed: [(COVID-19) AND (GST)], [(COVID-19) AND (glutathione S-transferases)], [(SARS-CoV-2) AND (GST)], [(SARS-CoV-2) AND (glutathione S-transferases)]. For long COVID, the following combinations were utilized: [(long COVID) AND (*GST*)], [(long COVID) AND (*glutathione S-transferases*)], and [(long COVID) AND (glutathione)] to obtain more information regarding the persistence of oxidative stress after the acute phase. Selection criteria: (a) Inclusion Criteria: Studies were included if they involved genotyping of *GST* family genes in patients with COVID-19 and long COVID, and explored the relationship between these gene polymorphisms and disease severity, mortality, or disease progression. In terms of long COVID, studies focused on the persistence of oxidative stress during the condition and the measurement of antioxidant enzymes. (b) Exclusion Criteria: Publications that were not in English, incomplete texts, theses, abstracts, case reports, or ecological studies were excluded.

Two reviewers, VVS and ACG, conducted risk of bias (RoB) assessments using the GRADE scale to evaluate the quality of the evidence [38]. We classified the included studies in accordance with the National Institute for Health and Care Excellence (NICE) evidence levels, which consider factors such as study design, methodology, validity, and applicability [39]. Studies rated as level 2+ included cohort or case-control studies that had a low risk of confounding, bias, or chance, and a moderate likelihood that the observed association is causal. Additionally, we applied the NICE grading system for recommendations to the studies, reflecting the degree of confidence that the estimated effect is strong enough to support a recommendation. Studies classified as grade D are based on extrapolations from studies rated as 2+.

In the process of preparing this systematic review, we utilized Grammarly^®^ (San Francisco, CA, USA) artificial intelligence (AI) exclusively for the purposes of stylistic editing and language refinement of the manuscript. These advanced tools played a crucial role in enhancing the clarity, coherence, and overall readability of the document. However, it is important to highlight that they did not contribute original content, data, or interpretations to our work. All significant intellectual contributions—such as the study design, data collection, analysis, and interpretation—were conducted independently by the authors. We took great care to ensure that the academic integrity and originality of our research remained intact. The use of AI was strictly confined to superficial aspects of text editing, including grammar corrections, punctuation adjustments, and stylistic improvements. Consequently, the application of these tools did not have any bearing on the scientific content, methodologies, or conclusions presented in this paper.

## 3. Results

Using our search strategy, we initially identified 154 articles related to COVID-19. After removing duplicates, 69 articles remained. An independent abstract screening excluded 55 articles that did not assess *GST* polymorphisms. A full-text review was conducted on the remaining 14 articles, of which 10 were excluded because of study design or non-relevant objectives. Ultimately, four studies investigating the role of *GST* polymorphisms in COVID-19 were included in this review.

For long COVID, 44 articles were initially identified. After removing duplicates, 41 articles remained. We excluded forty that did not evaluate *GST* polymorphisms, resulting in one article being included in this review. Figure 4 presents a flow diagram outlining the entire screening and selection process.

### 3.1. GST Polymorphisms and COVID-19 Studies

The relationship between *GST* polymorphisms and COVID-19 outcomes has been the subject of increasing scientific attention (Table 1), given the central role of *GSTs* in cellular detoxification and redox homeostasis. Among the *GST* family, *GSTM1* and *GSTT1* are the most studied because of their high degree of polymorphism, particularly the presence of null genotypes that result in reduced or absent enzymatic activity. This genetic variability has been hypothesized to influence susceptibility to oxidative stress-related pathologies, including those triggered by SARS-CoV-2 infection.

Several studies have explored the association between *GSTM1* and *GSTT1* null genotypes and COVID-19 severity, but the evidence remains inconclusive. In a cohort study by Mohammad Abbas et al. [40], which followed 269 Indian COVID-19 patients over one month, the frequencies of *GSTM1* and *GSTT1* null genotypes did not significantly differ between patients with mild and severe disease. The odds ratios for severe disease in carriers of *GSTM1*−/− and *GSTT1*−/− genotypes were not statistically significant, and similar results were observed when considering combinations of these genotypes. However, the study found that patients with the *GSTT1−/−* genotype had a significantly higher risk of death (HR 2.28, 95% CI 1.013–5.141; *p* = 0.047), and this risk was even higher in those with the *GSTM1+/+/GSTT1−/−* combination (HR 2.72, 95% CI 1.172–6.295; *p* = 0.02). No association between *GSTM1−/−* alone and mortality was observed.

Similarly, the case-control study by Orlewska et al. [41] involving both vaccinated and unvaccinated patients found no significant association between *GSTM1* or *GSTT1 null* genotypes and disease severity in either group. Vaccinated patients experienced significantly less severe disease overall, but the distribution of *GST null* genotypes was similar across severity categories and vaccination status. When the entire cohort was analyzed, neither *GSTM1−/−* nor *GSTT1−/−* was associated with increased disease severity. These findings were corroborated by Coric et al. [42], who found no significant association between *GSTM1* or *GSTT1 null* genotypes and susceptibility to COVID-19 in a case-control study including 207 COVID-19 patients and 252 controls.

Beyond *GSTM1* and *GSTT1*, other *GST* isoforms have also been investigated for their potential role in COVID-19 susceptibility and outcomes. Djukic et al. [43], reported that the *GSTO1 AA* genotype was associated with a higher likelihood of developing COVID-19 compared with *GSTO1* CC (OR 2.45, 95% CI 1.03–5.84, *p* = 0.044), and similar risk increases were observed for GSTO2 AG (OR 1.91, 95% CI 1.10–3.30, *p* = 0.020) and *GSTO2* GG (OR 3.69, 95% CI 1.62–8.40, *p* = 0.002) genotypes. Haplotype analysis confirmed a high degree of linkage disequilibrium between *GSTO1* and *GSTO2* polymorphisms, and the cumulative presence of risk genotypes further increased the probability of developing COVID-19. However, no significant association was found between these genotypes and disease severity. These results suggest a polygenic contribution to COVID-19 susceptibility, where the combined effect of multiple risk alleles may be more relevant than any single genotype.

The *GSTP1* isoform was also examined in the study by Orlewska et al. [41]. Among unvaccinated patients, neither the *GSTP1 Ile/Val* nor *Val/Val* genotypes were associated with disease severity. Interestingly, among vaccinated patients, those with the *GSTP1 Ile/Val* genotype had a significantly higher risk of developing moderate-to-severe disease (OR 2.74, 95% IC 1.05–7.18, *p* = 0.0398), suggesting that the effect of this genotype may be context-dependent and potentially influenced by immune status or other factors. However, these associations were not observed when the entire cohort was analyzed.

In the previously mentioned study by Coric [42] and collaborators from 2021, six genes were studied: *GSTP1, GSTM3, GSTM1, GSTT1,* and *GSTA1.* They found that carriers of the heterozygous *GSTP1 IleVal* (*rs1695*) genotype were less likely to develop COVID-19 (OR 0.66, 95% IC 0.44–0.98, *p* = 0.042), as were carriers of *GSTP1*Val* (*rs1138272*) allele in individuals carrying at least one Ala/Val allele (OR 0.63, 95% CI 0.41–0.99, *p* = 0.039), and the *Val/Val* genotype (OR 0.08, 95% CI 0.10–0.64, *p* = 0.017). For *GSTM3*, the *AC* genotype conferred a protective effect (OR 0.60, 95% CI 0.38–0.96, *p* = 0.033), while the *CC* genotype was associated with increased risk (OR 1.71, 95% CI 0.99–2.95, *p* = 0.053). The cumulative effect of risk genotypes across *GSTP1* and *GSTM3* was significant, with the risk of developing COVID-19 rising with each additional risk genotype. Notably, the presence of three risk genotypes was associated with a 4.7-fold increase in the risk of severe disease (OR 4.7, 95% CI 1.22–18.39, *p* = 0.025). These findings underscore the complexity of genetic susceptibility and highlight the importance of considering multiple loci and their interactions when assessing risk.

### 3.2. GST Polymorphisms and Long COVID Study

The relationship between *GST* polymorphisms and long COVID has also been explored (Table 1). Ercegovac et al. [44] followed a cohort of 167 patients for three months after acute illness to assess the antioxidant profile and neurological sequelae. A substantial proportion of patients experienced persistent symptoms such as fatigue, myalgias, and brain fog. The *GSTM1 null* genotype was associated with a more than two-fold increased risk of developing brain fog (OR 2.29, 95% CI 0.79–6.58), while carriers of at least one Val allele of *GSTP1 AB* (OR 0.43, 95% CI 0.21–0.87) and those with the *GSTO1 AspAsp* genotype (OR 0.51, 95% CI 0.25–1.02) had a reduced risk of developing myalgia. The combined effect of risk genotypes (*GSTP1 AB IleIle*, *GSTO1 AlaAla*, *GPX1 LeuLeu*, and *GPX3 CC*) markedly increased the probability of myalgia, suggesting a cumulative genetic burden.

## 4. Discussion

The studies presented indicate a varied relationship between *GST* polymorphisms and the development, severity, and mortality of COVID-19. Notably, the research conducted by Abbas [40] was the only one to reveal a statistically significant association between an increased risk of death and the *GSTT1−/−* genotype, as well as the combination of *GSTM1−/−* and *GSTT1+/+*. However, no significant association was found between these null genotypes and the clinical severity of the disease. This contrasts with findings reported by Orlewska et al. [41] and Coric et al. [42], who found no significant associations for either *GSTM1−/−* or *GSTT1*−/− concerning disease severity or development. Conversely, Djukic [43] reported a statistically significant increase in the risk of developing COVID-19 in individuals with specific *GSTO* polymorphisms, including the *GSTO1 AA*, *GSTO2 AG*, and *GSTO2 GG* genotypes. Similarly, Orlewska identified an increased risk associated with the *GSTM3 CC* genotype.

In contrast, some studies have indicated a protective effect that decreases the likelihood of developing the disease. For instance, Coric et al. [42] identified specific genetic variants of *GSTP1* that are associated with a lower risk of COVID-19. These include the *GSTP1 Ile/Val* genotype, the *GSTP1*Val allele* in individuals with at least one *Ala/Val allele*, and the *Val/Val* genotype. Additionally, individuals with the *GSTM3 AC* genotype also showed a reduced risk.

To date, only one study has investigated the association between these polymorphisms and the development of long COVID, particularly regarding neurological symptoms. Ercegovac et al. [44] found that individuals with the *GSTM1−/−* genotype were more likely to experience brain fog. Conversely, carriers of the *GSTP1 AB* and *GSTO1 AspAsp* genotypes appeared to have a potential protective effect.

This research primarily examines *GST* genetic variants and their potential link to long COVID. However, it is essential to recognize that the development of this condition is influenced by multiple factors and encompasses both the pre- and post-vaccination periods. In the study by Yonker et al. [45], elevated levels of free Spike protein (33.9 ± 22.4 pg/mL) were identified in adolescents who developed myocarditis following vaccination, with symptoms emerging within the first week after immunization. Supporting these findings, Craddock et al. [46] also found circulating Spike protein in 64% of patients with long COVID, along with circulating viral RNA in 59% of cases, persisting even 8 to 12 weeks after the initial infection. In comparison to healthy individuals, these levels showed a significant decline following recovery.

As previously highlighted, it is noteworthy that only one article has investigated the potential impact of polymorphisms on disease chronicity. However, compelling evidence from other researchers clearly demonstrates the persistence of oxidative stress (OS) in patients suffering from long COVID. Yang et al. [47] have definitively shown a decrease in superoxide dismutase (SOD) enzyme activity, accompanied by reduced levels of glutathione (GSH) and elevated levels of oxidized glutathione (GSSG) in patients whose chest CT scans indicate fibroproliferation. Consistently, Valente et al. [48] have reported similarly diminished levels of SOD and catalase (CAT). *GSTs* play a complementary role alongside superoxide dismutase (SOD) and catalase (CAT) within the antioxidant system, particularly in maintaining redox balance through glutathione metabolism. When the activities of SOD and CAT are compromised, as documented in the aforementioned studies, *GST* function becomes increasingly crucial in counteracting OS. In this context, polymorphisms in *GST* genes may reduce or abolish enzymatic activity, potentially contributing to the persistence of OS observed in patients with long COVID. Although direct evidence linking these genetic variants to long COVID remains limited, the functional interplay among GST, SOD, and CAT supports the hypothesis that *GST* polymorphisms may play a significant role in the progression and chronicity of the disease [47,48].

The findings indicate that specific polymorphisms in *GST* genes may influence mortality risk and susceptibility to both COVID-19 and long COVID. However, the impact of these polymorphisms on clinical severity remains inconsistent across various studies. Although not all findings achieved statistical significance, the observed trends and analyses regarding the cumulative effects of risk genotypes suggest a potential collective influence of multiple polymorphisms on disease susceptibility and severity. Additionally, four of the five studies were conducted in Europe, while one was conducted in Asia. This geographical concentration limits the genetic diversity represented, which is a limitation since allele frequencies of *GST* polymorphisms can vary among populations because of ethnic and genetic differences.

This review encompasses a variety of studies that employ different research designs, involve diverse populations, and reflect geographic variability, while also analyzing several polymorphisms in *GST*. This broadens the range of research significantly. However, several limitations must be acknowledged. Notably, there is considerable heterogeneity across the studies concerning the demographics of the populations examined and the criteria established to classify disease severity. This variability complicates direct comparisons between studies and diminishes the ability to generalize the findings to broader populations. Furthermore, some of the studies are based on relatively small sample sizes, which may compromise the statistical power of their analyses and potentially affect the reliability of the results. Last, many of the investigations do not succeed in establishing clear causal relationships between the *GST* polymorphisms studied and the clinical outcomes observed, leaving a gap in the understanding of how these genetic variations may influence disease progression or patient responses.

## 5. Conclusions

GST genes are important for maintaining a balance in the body and protecting against OS. This is especially relevant for COVID-19, which can cause oxidative damage. Most research focuses on cytosolic *GSTs* such as *GSTM1* and *GSTT1*, but the results are often inconsistent because of different study designs and genetic diversity. There is also not enough research on *GST* variations related to long COVID, particularly in Latin American populations. Including microsomal and mitochondrial GSTs in studies could provide new insights into how these enzymes affect OS and inflammation during viral infections. A deeper understanding of these enzymes could aid in developing targeted prevention and treatment strategies for COVID-19 and its long-term effects.

## Figures and Tables

**Figure 1 antioxidants-14-00912-f001:**
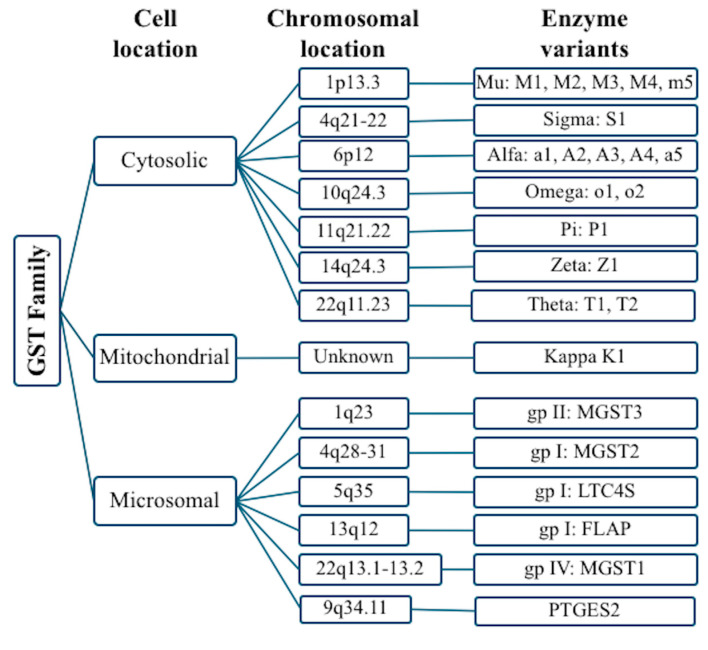
*Glutathione S-transferase* gene (GST) family. The *GST* genes are categorized into three distinct families based on their substrate specificity and the structure of their N-terminal domains.

**Figure 2 antioxidants-14-00912-f002:**
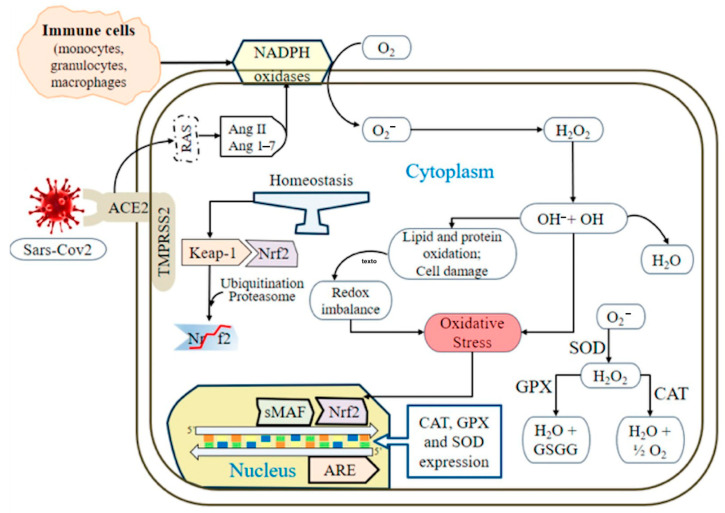
Overview of cellular ROS production and detoxification by antioxidants during SARS-CoV-2 infection. OS arises from the activity of angiotensin-converting enzyme (ACE), leading to an increased production of angiotensin II (Ang II). This process enhances the generation of reactive oxygen species (ROS) through NADPH oxidase and peroxynitrite anions. During SARS-CoV-2 infection, the virus decreases the levels of ACE2 on cell surfaces, resulting in the accumulation of toxic angiotensin II, which exacerbates inflammation and can lead to acute respiratory distress syndrome (ARDS). ROS production predominantly occurs via the mitochondrial electron transport chain and NADPH oxidases. In response to oxidative stress, nuclear factor erythroid 2-related factor 2 (Nrf2) and small MAF (sMAF) proteins activate antioxidant response elements (ARE), promoting the synthesis of antioxidant enzymes such as superoxide dismutase (SOD). Enzymes such as glutathione peroxidase (GPx) and catalase (CAT) help convert ROS into molecular water, facilitating detoxification.

**Figure 3 antioxidants-14-00912-f003:**
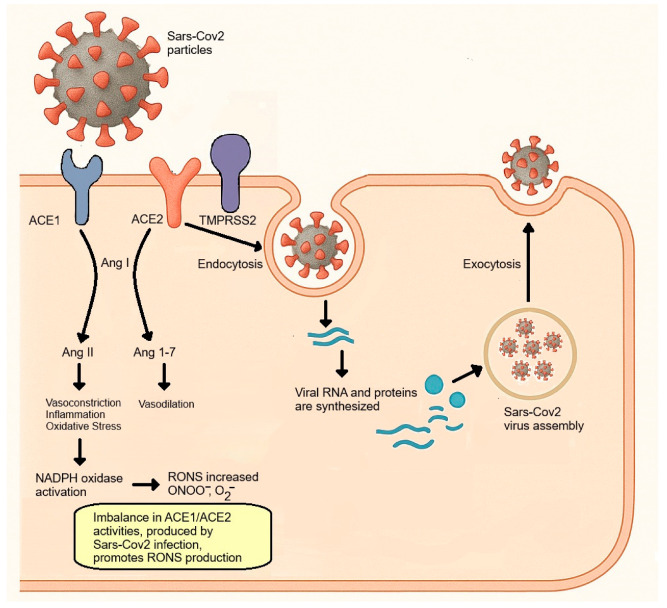
ACE2 dysfunction and oxidative stress pathways in SARS-CoV-2 Infection. SARS-CoV-2 infection disrupts the balance between the activities of ACE1 and ACE2, leading to increased oxidative stress. The virus binds to ACE2 and enters the cell through a process called endocytosis, which is facilitated by TMPRSS2. This interaction reduces ACE2 activity, causing an increase in levels of Ang II. Elevated Ang II promotes vasoconstriction, inflammation, and oxidative stress through the activation of NADPH oxidase and the increased production of RONS, such as peroxynitrite (ONOO^‒^) and O_2_^‒^. In a healthy state, ACE2 normally converts Ang I to Ang 1–7, which induces vasodilation. However, the imbalance created by SARS-CoV-2 infection favors oxidative stress and aids in viral replication and assembly within the host cell.

**Figure 4 antioxidants-14-00912-f004:**
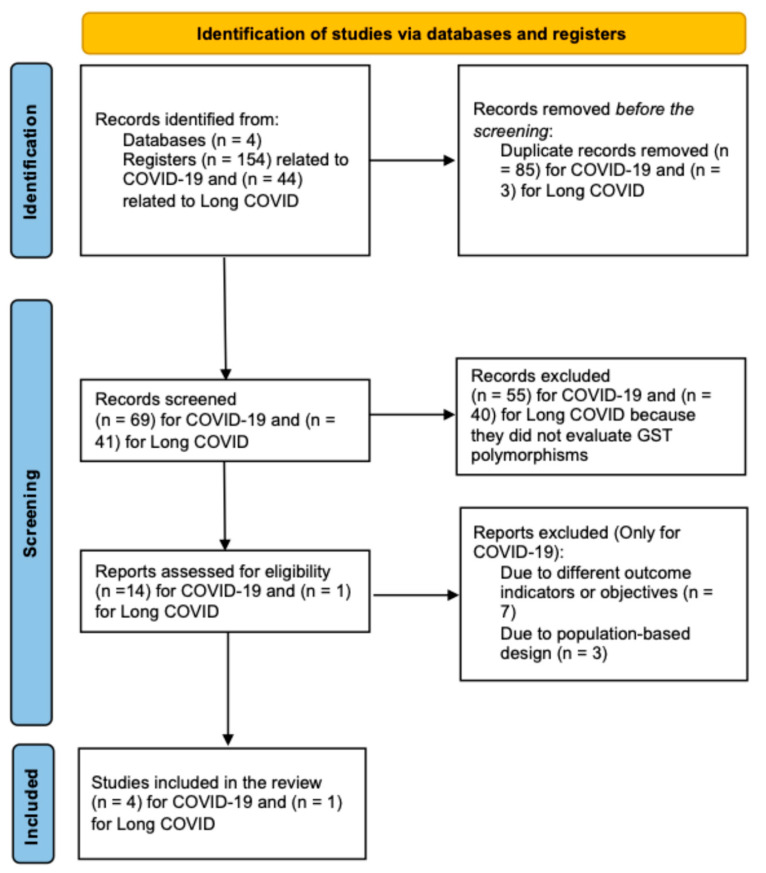
Algorithm for the selection of articles included in this study according to the systematic review methodology.

**Table 1 antioxidants-14-00912-t001:** Description of the relationships between the presence of GST polymorphisms and severity, development, or mortality in patients with COVID-19, and long COVID.

Author	Year	Study Design	Country	Disease	Sample	Studied Genes	Results	Evidence Level	Grade of Recommendation	Reference
Abbas, M	2021	Cohort	India	COVID-19	269 patients with COVID-19 confirmed by RT-PCR with mild (n = 149) and severe (n = 120) infection	*GSTM1/GSTT1*	Increased risk of death in *GSTT1−/−* vs. *GSTT1+/+* (HR 2.28, 95% CI 1.013–5.141; *p* = 0.047) and in *GSTM1+/+/GSTT1−/−* (HR 2.72, 95% CI 1.172–6.295; *p* = 0.02).	2+ Low certainty	D	[40]
Orlewska, K	2023	Cases and controls	Poland	COVID-19	176 patients with confirmed SARS-CoV-2 infection.	*GSTM1/GSTT1/GSTP1*	Vaccinated patients carrying *GSTP1 Ile/Val* were 2.74 times more likely to develop a moderate-to-severe form of the disease (OR 2.74, 95% CI 1.05–7.18, *p* = 0.0398).	2+ Low certainty	D	[41]
Coric, V	2021	Cases and controls	Serbia	COVID-19	459 Caucasian patients: 207 with RT-PCR-confirmed COVID-19 > 18 years and 252 without IgG and IgM antibodies for SARS-CoV-2.	*GSTP1/GSTM3/GSTM1/GSTT1/GSTA1*	Lower risk of developing the disease GSTP1 IleVal rs1695 (OR 0.66, 95% CI 0.44–0.98, *p* = 0.042). *GSTP1*Val rs1138272* with at least one allele (*AlaVal* OR 0.63, *p* = 0.039; *ValVal* OR 0.08, *p* = 0.017) vs. wild type allele. *GSTM3 AC* compared with *GSTM3 AA* (OR 0.60, 95% CI 0.38–0.96, *p* = 0.033). Higher risk of developing severe disease *GSTM3 CC* vs. *GSTM3 AA* (OR 2.5, 95% CI 1.13–5.61, *p* = 0.024). The cumulative effect of two risk genotypes (OR 3.38, 95% CI 1.56–7.34, *p* = 0.002) For three genotypes (OR 11.86, 95% CI 2.84–49.40, *p* = 0.001)	2+ Low certainty	D	[42]
Djukic, T	2022	Cases and controls	Serbia	COVID-19	491 Caucasian patients: 255 with COVID-19 confirmed by RT-PCR > 18 years and 236 without IgG and IgM antibodies for SARS-CoV-2.	*GSTO1, GSTO2*	Higher risk of developing disease *GSTO1*AA* vs. *GSTO1*CC* genotype (OR = 2.45, 95% CI 1.03–5.84, *p* = 0.044). *GSTO2*AG* genotype (OR = 1.91, 95% CI 1.10–3.30, *p* = 0.020). *GSTO2*GG* genotype (OR = 3.69, 95% CI 1.62–8.40, *p* = 0.002). The haplotype H2: *GSTO1*A* (rs4925) and *GSTO2*G* (*rs15669*) (OR = 1.97, 95% CI: 1.28–3.03, *p* = 0.002).	2+ Low certainty	D	[43]
Ercegovac, M	2022	Cohort	Serbia	Long COVID	167 patients recruited 3 months after COVID-19	*GSTM1/GSTT1/GSTO1/GSTP1*	Lower risk of developing myalgia Carriers of at least one Val allele of the *GSTP1AB* (OR = 0.43, 95% CI: 0.21–0.87). The combined *GSTP1AB ValVal* and *GSTO1 AspAsp* genotypes (OR = 0.24, 95% CI: 0.09–0.66) compared with the combined *GSTP1 IleIle* and *GSTO1 AlaAla* genotypes. Higher risk of developing myalgia The cumulative effect of the *GSTP1 AB IleIle*, *GSTO1 AlaAla*, *GPX1 LeuLeu*, and *GPX3 CC* (OR = 10.5, 95% CI: 1.67–66.09).	2+ Low certainty	D	[44]

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
