# Peer review of "A Systematic Review of Genetic Variants in Glutathione S-Transferase Genes and Their Dual Role in SARS-CoV-2 Pathogenesis: From Acute Respiratory Complications to Long COVID"

_antioxidants, 2025, doi:10.3390/antiox14080912_

Round 1
Reviewer 1 Report
The paper by Villegas Sànchez try to focus on genetic variants of GSTs and SARS-CoV-2 pathogenesis.
The topic is interesting but there are some concerns.
Major comments
(1) I have a concern about the studies reported in Table 2. The authors state that those studies were selected because focused on GSTs polymorphisms. However - as judged by the results reported - only the paper by Ercegovac (42) fouses on GSTs polymorphisms (see the column “Results”). I mean, it is certainly allowed and useful to describe the biomarkers of oxidative stress in patients with long COVID, but what the reader expects from the title of this review is rather know something about the role of GSTs polymorphisms in long COVID.
Of course, there are connections between GST, GSH/GSSG levels and oxidative stress markers, but the focus must remain on GSTs polymorphisms, otherwise the review is reporting something other than GST polymorphisms (namely, oxidative stress in the pathogenesis of SARS-CoV-2). In the latter case, the title and the first part of the review (talking about acute respiratory complications in COVID and GSTs polymorphisms) should be changed and the review focused on "oxidative stress and COVID".
In other words, the review seems to start with one aim but ends with another. If there are no manuscripts that - according to the criteria chosen by the authors - study the polymorphisms of the GSTs in long COVID (with the exception of ref. 42) the authors can also omit this section or add it to the first part of the review.
(2) Another concern is that GSH, GSSG, AGEs etc that are indicated under the title “Studied genes” (Table 2) are molecules, not genes…
(3) Another major concern is about the “Discussion” section. What is reported from line 277 to line 348 is more to be considered as part of the "Results" section, rather than a reasoned discussion of the results obtained.
There is no real discussion of the results obtained. There is certainly a general comment on the limited number of studies focusing on GSTs polymorphisms and COVID and the lack of clear results (“…the results have frequently been inconsistent and at times contradictory”). However, does it depend on a low number of studies? What are the polymorphisms of the GSTs for which correlations are found? Why? What are their specificities (function, substrates, etc.)? Why do they have a "dual role" (see the title)?
(1) Figure 1 should be updated considering that also prostaglandin E synthase (PGES) belongs to the MAPEG (membrane-associated proteins in eicosanoid and glutathione metabolism) superfamily.
(2) Page 2 – lines 85-90 – The same concepts are repeated twice.
(3) Page 3, line 101 – “least”. Maybe “less”?
Author Response
Dear Editor: Below are the point-by-point responses to the reviewer 1 of the article entitled:
A Systematic Review of Genetic Variants in Glutathione S-transferase Genes and Their Dual Role in SARS-CoV-2 Pathogenesis: From Acute Respiratory Complications to Long COVID. (Antioxidants-3687759)
Reviewer 1
There are some concerns. 1) Check the language used in Figure 4 and table 2 (Ref. 44). 2) Table 1 and 2 – What do you mean with “Evidence level” and “Grade of recommendation”? It should be explained to what the Authors are referring to, the meaning of the various grades and levels, etc 3) The formatting of tables should be revised. It is not clear where the results of a study end and where the following study begin ("Results" column). 4) The text of the tables should be abbreviated and broken down into points as if it were a list. A text that is too long fits better into the "Results" section of the manuscript.
Observation 1):
- Check the language used in Figure 4 and table 2 (Ref. 44).
Response 1:
We appreciate your comments. We have made appropriate modifications to the language used in Figure 4 and have removed Table 2 as part of the adjustments made in response to Major Comment 1 (below).
Observation 2) :
- Table 1 and 2 – What do you mean with “Evidence level” and “Grade of recommendation”? It should be explained to what the Authors are referring to, the meaning of the various grades and levels, etc
Response 2:
The GRADE methodology was used to evaluate the quality of the evidence, which allows us to systematically and standardizedly establish both the quality of the evidence and the level of certainty of the findings. Additionally, the included studies were categorized according to the grades of recommendation defined by the NICE guidelines, so two references were added on which we based our classification (DOI: 10.1016/j.jclinepi.2010.07.015, doi: 10.4067/S0716-10182014000600011.) The changes were made in lines 158-165 in the Methodology section.
Observation 3) :
- The formatting of tables should be revised. It is not clear where the results of a study end and where the following study begin ("Results" column).
Response 3:
Table 2 has been deleted due to the request in major comment 1 (below), and the format of Table 1 has been modified to make the results section easier to read. Only one of the items in Table 2 was included in Table 1, and the “Disease” column was added to differentiate the COVID-19 items from the long COVID items.
- The text of the tables should be abbreviated and broken down into points as if it were a list. A text that is too long fits better into the "Results" section of the manuscript.
Response 4:
We are very grateful for your feedback. We have reorganized the text of Table 1 into bullets and abbreviated the results.
Observation 2.1:
Major comments
(1) I have a concern about the studies reported in Table 2. The authors state that those studies were selected because focused on GSTs polymorphisms. However - as judged by the results reported - only the paper by Ercegovac (42) fouses on GSTs polymorphisms (see the column “Results”). I mean, it is certainly allowed and useful to describe the biomarkers of oxidative stress in patients with long COVID, but what the reader expects from the title of this review is rather know something about the role of GSTs polymorphisms in long COVID.
Of course, there are connections between GST, GSH/GSSG levels and oxidative stress markers, but the focus must remain on GSTs polymorphisms, otherwise the review is reporting something other than GST polymorphisms (namely, oxidative stress in the pathogenesis of SARS-CoV-2). In the latter case, the title and the first part of the review (talking about acute respiratory complications in COVID and GSTs polymorphisms) should be changed and the review focused on "oxidative stress and COVID".
In other words, the review seems to start with one aim but ends with another. If there are no manuscripts that - according to the criteria chosen by the authors - study the polymorphisms of the GSTs in long COVID (with the exception of ref. 42) the authors can also omit this section or add it to the first part of the review.
Response 2.1:
We thank you for your comments. We have eliminated studies focused on oxidative stress biomarkers in Long COVID to keep the focus of the article consistent. Because of this, we have adjusted the literature search for Long COVID and GST polymorphisms exclusively in the methodology part. This resulted in adjusting Figure 4, deleting Table 2 and modifying Table 1.
Observation 2.2:
Another concern is that GSH, GSSG, AGEs etc that are indicated under the title “Studied genes” (Table 2) are molecules, not genes…
Response 2.2:
We appreciate your suggestion on the title of Table 2. According to your kind remark, table 2 has been deleted and the Long COVID related item has been added to table 1, which includes GST genetic variant only.
Observation 2.3:
(3) Another major concern is about the “Discussion” section. What is reported from line 277 to line 348 is more to be considered as part of the "Results" section, rather than a reasoned discussion of the results obtained.
Response 2.3
There is no real discussion of the results obtained. There is certainly a general comment on the limited number of studies focusing on GSTs polymorphisms and COVID and the lack of clear results (“…the results have frequently been inconsistent and at times contradictory”). However, does it depend on a low number of studies? What are the polymorphisms of the GSTs for which correlations are found? Why? What are their specificities (function, substrates, etc.)? Why do they have a "dual role" (see the title)?
We appreciate your comment, which we consider highly relevant for clarifying the questions you have raised. Accordingly, we have revised and reorganized the Discussion section, adding explicit and more detailed information regarding the main findings and their relevance. We have also included a discussion of the study's strengths and limitations and emphasized the importance of these results. The available evidence is based on a limited number of studies, which introduces heterogeneity and limits the ability to establish definitive associations between GST polymorphisms and COVID-19 outcomes. We also include a paragraph in the 338-349 lines indicating that these genetic variants appear to play a dual role, influencing both the acute and chronic phases of the disease. Since these enzymes play a fundamental role in cellular detoxification and maintaining redox balance, polymorphisms in their genes may alter enzymatic function. This alteration could contribute to increased oxidative stress during the acute phase and its prolonged persistence, thus representing a potential risk factor for the development of Long COVID.
Detailed comments
Observation 1.1:
(1) Figure 1 should be updated considering that also prostaglandin E synthase (PGES) belongs to the MAPEG (membrane-associated proteins in eicosanoid and glutathione metabolism) superfamily.
Response 1.1
Thank you for your valuable observation. We have updated Figure 1 to include prostaglandin E synthase (PGES) as a member of the MAPEG (membrane-associated proteins in eicosanoid and glutathione metabolism) superfamily, as suggested. A search was conducted in PUBMED to determine the subfamily to which it belongs; however, no relevant articles were found. As a result, it was included in the figure, but without specifying its subfamily classification.
Observation 2.1:
(2) Page 2 – lines 85-90 – The same concepts are repeated twice.
Response 2.1
We appreciate your kind correction. We have condensed the information in that paragraph from lines 68 to 77 to reduce redundancy.
Observation 3.1:
(3) Page 3, line 101 – “least”. Maybe “less”?
Response 3.1
Thank you for your insightful observation. However, we are uncertain whether you are referring to the paragraph that contains the word "least" in line 111: “A complication of this disease is Long COVID or post-COVID syndrome, which involves the persistence or emergence of new symptoms three months after infection, lasting at least two months without an alternative explanation [29][30].” If that is the case, we want to clarify that we reviewed the paragraph using the Grammarly style checker and also had it read by a Native American colleague, who indicated that it is error-free. Therefore, we believe the paragraph is written correctly. Nevertheless, we remain open to any further suggestions you may have.

Reviewer 2 Report
This systematic review article examines the correlation between types of Glutathione S-transferase genetic variants and the pathogenicity of SARS-CoV-2. It is a synthesis of work carried out in this field between 2020 and 2025. It may also be useful to illustrate that the capacity to resist oxidative stress is associated with the severity of the disease and its complications, such as long covid. The introduction is useful because it shows the relationship between virus binding to the ACE2 receptor and the generation of oxidative stress. This interaction is possible because the Spike protein is a ligand for the ACE2 receptor.
However, it appears that spike protein from either virus (Craddock V 2023 https://doi.org/10.1002/jmv.28568) or vaccine (Yonker LM 2024 https://doi.org/10.1161/CIRCULATIONAHA.122.061025) origin may persist. It would be useful to mention this and, if possible, put it into the context of this review, which covers two periods, pre- and post-vaccination. This parameter could be included in the discussion in the section between lines 304 and 309.
Author Response
Dear Editor: Below are the point-by-point responses to the reviewer 2 of the article entitled:
A Systematic Review of Genetic Variants in Glutathione S-transferase Genes and Their Dual Role in SARS-CoV-2 Pathogenesis: From Acute Respiratory Complications to Long COVID. (Antioxidants-3687759)
Reviewer 2
Observation 1:
This systematic review article examines the correlation between types of Glutathione S-transferase genetic variants and the pathogenicity of SARS-CoV-2. It is a synthesis of work carried out in this field between 2020 and 2025. It may also be useful to illustrate that the capacity to resist oxidative stress is associated with the severity of the disease and its complications, such as long covid. The introduction is useful because it shows the relationship between virus binding to the ACE2 receptor and the generation of oxidative stress. This interaction is possible because the Spike protein is a ligand for the ACE2 receptor.
Detailed comments
However, it appears that spike protein from either virus (Craddock V 2023 https://doi.org/10.1002/jmv.28568) or vaccine (Yonker LM 2024 https://doi.org/10.1161/CIRCULATIONAHA.122.061025) origin may persist. It would be useful to mention this and, if possible, put it into the context of this review, which covers two periods, pre- and post-vaccination. This parameter could be included in the discussion in the section between lines 304 and 309.
Response 1:
We appreciate your observation regarding the complexities of Long COVID. It is crucial to underscore the multifactorial nature of this condition, which involves a range of contributing factors that extend beyond genetic predispositions. While we acknowledge that GST polymorphisms represent a notable risk factor, it is important to clarify that they should not be viewed as definitive predictors of disease onset or severity. In the fourth paragraph of the discussion (lines 327-335), we have included other factors that influence the development of Long COVID, citing the recommended bibliography.

Round 2
Reviewer 1 Report
The authors replied to all the concerns raised by the reviewer.
However I have a concern regarding what stated in lines 338-349. Long COVID is associated with oxidative stress, reduced SOD and CAT activities and an imbalance of redox status. OK, but how can this lead to the conclusion that "Thus, it is clear that genetic variants in glutathione S-transferase (GST) play a significant role in the chronicity of the disease"? On what basis?
The limited number of existing studies on this (now very) specific topic (n. 5, with 3 studies coming from the same group) makes the manuscript more suitable for the format of a minireview, rather than a "full review".
Moreover, the same geographical origin of the subjects of 3 out of 5 available studies (Serbia - Europe) also has the limit of the low variability of the polymorphisms investigated (whose incidence changes in different populations). This is also something that should be pointed out in the discussion.
None.
Author Response
Dear Editor: Below are the point-by-point responses to the reviewer 1 of the article entitled:
A Systematic Review of Genetic Variants in Glutathione S-transferase Genes and Their Dual Role in SARS-CoV-2 Pathogenesis: From Acute Respiratory Complications to Long COVID. (Antioxidants-3687759)
- Observation 1
However I have a concern regarding what stated in lines 338-349. Long COVID is associated with oxidative stress, reduced SOD and CAT activities and an imbalance of redox status. OK, but how can this lead to the conclusion that "Thus, it is clear that genetic variants in glutathione S-transferase (GST) play a significant role in the chronicity of the disease"? On what basis?
Response 1:
We thank you for your comments. GSTs have a close functional relationship with other antioxidant enzymes, such as SOD and CAT, as all are part of the glutathione-based antioxidant system. As reported by Yang and Valente, the activity of these enzymes is diminished in patients with Long COVID, making the glutathione system even more critical. Polymorphisms in GST genes can impair their enzymatic activity, reducing the overall detoxification capacity and contributing to persistent oxidative stress. This redox imbalance may play a role in the chronicity of the disease. Although only one study has directly explored the association between GST polymorphisms and disease chronicity, the central role of GST in the antioxidant defense system, together with consistent evidence of sustained oxidative stress, supports this hypothesis from a pathophysiological standpoint. The text in lines 338 to 355 has been revised to offer a more comprehensive rationale and better explain the connection between GST polymorphisms and Long COVID.
Paragraph added:
“As previously highlighted, it is noteworthy that only one article has investigated the potential impact of polymorphisms on disease chronicity. However, compelling evidence from other researchers clearly demonstrates the persistence of oxidative stress (OS) in patients suffering from Long COVID. Yang et al. [47] have definitively shown a decrease in superoxide dismutase (SOD) enzyme activity, accompanied by reduced levels of glutathione (GSH) and elevated levels of oxidized glutathione (GSSG) in patients whose chest CT scans indicate fibroproliferation. Consistently, Valente et al. [48] have reported similarly diminished levels of SOD and catalase (CAT). GSTs play a complementary role alongside superoxide dismutase (SOD) and catalase (CAT) within the antioxidant system, particularly in maintaining redox balance through glutathione metabolism. When the activities of SOD and CAT are compromised, as documented in the aforementioned studies, GST function becomes increasingly crucial in counteracting oxidative stress (OS). In this context, polymorphisms in GST genes may reduce or abolish enzymatic activity, potentially contributing to the persistence of OS observed in patients with Long COVID. Although direct evidence linking these genetic variants to Long COVID remains limited, the functional interplay among GST, SOD, and CAT supports the hypothesis that GST polymorphisms may play a significant role in the progression and chronicity of the disease [47], [48].”
Observation 2
The limited number of existing studies on this (now very) specific topic (n. 5, with 3 studies coming from the same group) makes the manuscript more suitable for the format of a minireview, rather than a "full review”.
Response 2:
Thank you for your comment. We agree with your point of view; considering the length of the proposed article, the limited number of studies published to date, the topicality of the bibliography, and the specificity of the topic addressed, it is better suited to a mini-review format. Therefore, we leave this decision to you and the editor-in-chief for your consideration.
- Observation 3
Moreover, the same geographical origin of the subjects of 3 out of 5 available studies (Serbia - Europe) also has the limit of the low variability of the polymorphisms investigated (whose incidence changes in different populations). This is also something that should be pointed out in the discussion.
Response 3:
We greatly appreciate your insightful observation that geographic and ethnic diversity is one of the main limitations of the proposed study. We have added a paragraph on lines 363 to 367 that emphasizes this limitation and discusses its impact on the genetic representativeness and variability of the analyzed polymorphisms.
Paragraph added:
“Additionally, four out of the five studies were conducted in Europe, while one was conducted in Asia. This geographical concentration limits the genetic diversity represented, which is a limitation since allele frequencies of GST polymorphisms can vary among populations due to ethnic and genetic differences.”